# Pin-on-Disc Modelling with Mesh Deformation Using Discrete Element Method

**DOI:** 10.3390/ma15051813

**Published:** 2022-02-28

**Authors:** Yunpeng Yan, Rudy Helmons, Dingena Schott

**Affiliations:** 1Department of Maritime and Transport Technology, Delft University of Technology, 2628 CD Delft, The Netherlands; r.l.j.helmons@tudelft.nl (R.H.); d.l.schott@tudelft.nl (D.S.); 2Department of Mineral Production & HSE, Norwegian University of Science and Technology, 7031 Trondheim, Norway

**Keywords:** sliding wear, mesh deformation, calibration of wear coefficient, scaling factor

## Abstract

The pin-on-disc test is a standard sliding wear test used to analyse sliding properties, including wear contour and wear volume. In this study, long-term laboratory test performance is compared with a short-term numerical model. A discrete element method (DEM) approach combined with an Archard wear model and a deformable geometry technique is used. The effect of mesh size on wear results is evaluated, and a scaling factor is defined to relate the number of revolutions between the experiment and the numerical model. The simulation results indicate that the mesh size of the disc has a significant effect on the wear contour. The wear depth and wear width follow a normal distribution after experiencing a run-in phase, while the wear volume has a quadratic relation with the number of revolutions. For the studied material combination, the calibration of the wear coefficient shows that the wear volume of the pin-on-disc test accurately matches the simulation results for a minimum of eight revolutions with a wear coefficient lower than 2 × 10^−11^ Pa^−1^.

## 1. Introduction

Bulk solids handling plays a significant role in a range of industries, such as the mining, agricultural, chemical and pharmacology industries [1]. For the mining industry, the process of transferring bulk solids, e.g., iron ore, leads to surface wear of handling equipment. Studies show that approximately 82% of the energy loss is attributed to the bulk material sliding along the chute bottom and 9% of the losses due to sliding against the side walls [2]. The sliding wear can be characterised as a relative motion between two solid surfaces in contact under load [3], and long-term wear leads to surface deformation and accelerates the damage of the equipment, resulting in a reduction in lifespan. To reduce the sliding wear of the surfaces of bulk solids handling equipment, a convex pattern surface is proposed [4] and optimised [5] by using the discrete element method (DEM) [6]. However, the deformation of the surface caused by the sliding wear is still unstudied. It is essential to investigate the deformation behaviour of the surface caused by the sliding wear because surface deformation might affect the flow, in turn influencing the wear behaviour of the geometry. Before the analysis of the deformation of the surface caused by bulk material, it is necessary to achieve the modelling of the surface deformation caused by a single particle, so a pin-on-disc test, which is a standard sliding wear test, is applied to the analysis of the surface deformation [3].

In the context of wear evaluation, DEM is a useful approach to predict the wear of equipment caused by bulk material. DEM was developed by Cundall and Strack to model particle systems by tracking the movement of each particle and its interaction with its surroundings over time [6], and it is widely used to design, analyse and optimise bulk material handling systems and equipment for granular materials [7].

On one hand, DEM simulates the wear process without the consideration of geometrical deformation. Cleary et al. [8] first proposed an approach based on DEM to predict the liner wear and distribution of a ball mill using a 2D model for different conditions of rotational velocity. Similarly, Cleary et al. [9,10] evaluated wear in a 3D slice of a mill and performed an evaluation and comparison between wear in tower and pin mills. Recently, Xu et al. [11,12,13] studied the liner wear of a tumbling mill based on a multiple-level approach, and the numerical model was validated by experimental data. Rojas et al. [14] studied the wear in mining hoppers and obtained comparable results with the measurements. In addition, Kalácska et al. [15], Katinas et al. [16] and Powell et al. [17] studied the wear behaviour of steel used in agricultural tines, wear of soil rapper tin, high speed steel and liner revolution in ball mills based on the discrete element method, respectively. On the other hand, some researchers are interested in surface deformation caused by wear. Kalala et al. [18] applied DEM to estimate adhesion, abrasion and impact wear in dry ball mills with further validation with industrial wear measurement. Esteves et al. [19] compared the industrial vertical stirred mills screw liner wear profile to the measurements after more than 3000 h by using scaling-up procedures for rotational velocities. The predicted wear volumes obtained from the DEM model have a good agreement with the measured results when using a specific velocity. Boemer et al. [20] proposed a generic wear prediction procedure based on the discrete element method for ball mill liners in the cement industry. By obtaining a global wear constant and analysing the mesh size sensitivity, the predicted wear profile can match the measurement through a mesh smoothing technique. Additionally, Schramm et al. [21] modelled a scratch test to study the abrasive material loss at soil tillage and compared it with a cross-section profile.

Although these studies pay attention to the deformation of geometry, the detailed analyses, such as mesh size sensitivity, wear contour matching and wear depth distribution, are still lacking, and these aspects determine the accuracy of the numerical model. Therefore, a pin-on-disc test as a benchmark is applied to this study to compare the long-term laboratory test with a short-term numerical model. Four steps are built to complete the research. First, the geometrical deformation technique is introduced, combined with a sliding wear model named the Archard wear model. Second, the procedures and the results of the pin-on-disc test are clarified. Third, the wear coefficient is calibrated, including the mesh size sensitivity, wear depth and wear width distribution analysis and wear contour reconstruction. Fourth, the simulation results including wear contour and wear volume analysis are verified by the test results.

## 2. Materials and Methods

### 2.1. Discrete Element Method

The motion of discrete spheres in DEM is governed by Newton’s second law of motion [7] as shown in Figure 1. Hertz–Mindlin no-slip contact model is a nonlinear elastic contact model which is appropriate for non-cohesive granular materials. This model consists of two springs, two dampers and a slider. The springs are used to represent particle stiffness in normal and tangential directions. Two dampers are used to model the damping forces, and the slider is applied to generate a friction force. The normal force $F_{n}(N)$ is calculated according to Equation (1), where $S_{n}(N/m^{3/2})$, $\delta_{n}(m)$, $D_{n}(N\cdot s/m)$, and $v_{n}(m/s)$ are the stiffness, overlap, damping coefficient and velocity in the normal direction of the contact, respectively.
(1)$$F_{n}=-\frac{2}{3}S_{n}\delta_{n}^{\frac{3}{2}}+D_{n}v_{n}$$

The tangential force $F_{t}(N)$ is restrained by Coulomb law [22], which is expressed by Equation (2), where $\mu_{st}$ is the coefficient of static friction; $S_{t}(N/m)$, $\delta_{t}(m)$, $D_{t}(N\cdot s/m)$, and $v_{t}(m/s)$ are the stiffness, overlap and coefficient of damping force and velocity in the tangential direction of the contact, respectively.
(2)$$F_{t}=\min\{-S_{t}\delta_{t}+D_{t}v_{t},\mu_{st}F_{n}\}$$

### 2.2. Geometrical Plastic Deformation Technique

In EDEM [23], the geometrical plastic deformation technique is related to the Archard wear model [24] based on a mesh deformation approach [25]. The Archard wear model is widely used both for the particles and geometries [14,16,17,19,20,25,26,27], such as the evaluation of the wear of ballast and abrasive grain [26,27], the prediction of the wear of mill lifters [28] and local failure prediction of abrasive wear on tipper bodies [29]. The Archard wear model can be denoted as Equation (3).
(3)$$V=k\frac{F_{n}}{H_{s}}l_{s}$$
where V($mm^{3}$) is the wear volume, $H_{s}$ ($N/mm^{2}$) is the hardness of the surface, k is a dimensionless wear coefficient, $F_{n}$(N) is the normal force applied to an equipment surface and $l_{s}$(mm) is the sliding distance.

Equation (3) can be expressed by a derivative formula denoted as Equation (4).
(4)$$dV=\frac{kF_{n}\left\|v_{t}\right\|dt}{H_{s}}=\alpha_{s}F_{n}\left\|v_{t}\right\|dt$$
where $\alpha_{s}=k/H_{s}$ represents the wear coefficient. dV,$\left\|v_{t}\right\|$ and dt denote the increment of wear volume of the material removed, relative tangential velocity and time increment, respectively.

The disc is meshed by triangular meshes with specific size using the software ANSYS Workbench [30] and the sequence of how the mesh is deformed is illustrated in Figure 2. First, the contact between the particle and the mesh elements is detected and the forces are calculated based on the contact model. Second, the element is displaced in normal direction of the element and the loss of the material is evaluated based on the Archard wear model. Third, the new positions of the elements and speed of the particle are recalculated. Therefore, the representation of the wear loss is performed by deforming the triangular meshes subjected to abrasive wear.

The wear volume of the $i^{th}$ element at each time step Δt is expressed as Equation (5).
(5)$$\Delta V_{i}=\int_{t}^{t+h}\alpha_{s}F_{n}\left\|v_{t}\right\|dt\approx \alpha_{s}F_{n}\left\|v_{t}\right\|\Delta t$$

The differential displacement (wear depth) $\Delta d_{i}$ for the $i^{th}$ element is related to the element area $A_{i}$ and the relations are denoted as Equations (6) and (7).
(6)$$\Delta d_{i}=\frac{\Delta V_{i}}{A_{i}}$$
(7)$$A_{i}=\frac{\left|\left(p_{1}-p_{2})\times (p_{1}-p_{3}\right)\right|}{2}$$
where $p_{1}$, $p_{2}$, $p_{3}$ are the nodes of the triangular element.

The wear depth $d_{i}^{t+\Delta t}$ and the new position of nodes of the mesh element $p_{k}^{t+\Delta t}$ at time t+Δt are obtained as Equations (8) and (9).
(8)$$d_{i}^{t+\Delta t}=d_{i}^{t}+\Delta d_{i}^{t}\cdot {\hat{n}}_{i}$$
(9)$$p_{k}^{t+\Delta t}=p_{k}^{t}+\Delta d_{i}^{t}\cdot {\hat{n}}_{i}\left(k=1,2,3\right)$$
where $\hat{n_{i}}$ is the normal vector of the mesh element. By interconnecting the common nodes of the element faces, it is possible to obtain continuity in the deformation of the surface, generating a smoothed wear pattern [25]; this phenomenon can be called a self-smoothing effect.

As the mesh element is in motion, the central point $p_{c}$, which represents the position of the ith element, is calculated before obtaining the accumulated wear $V_{i}$ of the mesh element as expressed by Equation (10).
(10)$$p_{ci}=\frac{p_{1,i}+p_{2,i}+p_{3,i}}{3}$$

Then, the wear volume of each element is integrated as Equation (11).
(11)$$V_{i}=V_{i}+\left|p_{cj}^{t+h}-p_{cj}^{t}\right|A_{i}$$

The total volume loss of the whole surface is obtained by adding the volume of each element together, denoted as Equation (12).
(12)$$V=\sum_{i}^{n}V_{i}$$

### 2.3. Pin-on-Disc Test Setup

Figure 3 illustrates a pin-on-disc tribometer [4] that is used to obtain sliding wear loss of a sample by single particle. The device consists of three main parts: the load which generates normal force over the pin, the pin mounted in a holder and a disc sample which suffers the sliding wear. The pin is located at a position with a distance of 22 mm from the centre of the disc, and the radius of which is 40 mm.

The materials of the disc and the pin are mild steel and iron ore, respectively, and the parameters of the test are listed in Table 1. The mild steel contains up to 0.3% carbon. During particle indentation, the disc surface deforms when the hardness ratio ($H_{p}/H_{s}$) of particle to the surface is higher than 1.2 and approximately maintains a constant when the ratio is higher than 1.9 [32]. Based on a statistical analysis of the hardness [4], the Vickers hardness of the particle and mild steel are 476±9$H_{v}$ and 143±4$H_{v}$, respectively, so the ratio is higher than 1.9 and therefore the disc is considered to be deformed with a relatively constant rate. As listed in Table 1, the rotational speed is 390.8 deg/s under the load of 5*N*, so the corresponding sliding distance after 1302.5 revolutions is 180 m.

## 3. Pin-on-Disc Test

Figure 4 illustrates the prepared particle, worn disc and obtained wear contours. The polished particle fixed in a metal holder as shown in Figure 4a is mounted to the tribometer. Figure 4b displays an example of the cross-sectional wear morphology on the mild steel disc after test. During the sliding process, grooves are formed as a result of the removal and displacement of the mild steel. To analyse the wear profile specifically, Figure 4c demonstrates three inhomogeneous wear morphologies obtained from three random positions of the wear path. It can be seen that the wear depth and width are 0.01–0.014 mm and 0.65–0.8 mm, respectively. The wear volume obtained is 0.565 ± 0.089 mm and the corresponding wear coefficient is (6.3 ± 1) × 10^−13^ Pa^−1^.

It should be noted that the main mechanisms of the sliding wear on the disc include micro-ploughing and micro-cutting, as mild steel is a ductile material [32]. The micro-ploughing generates ridges of deformed material which are pushed along ahead of the particle, as shown in Figure 4c. The micro-cutting deflects the material which flows up the front face of the particle to form a chip, so all the material displaced by the particle is removed in the form of chips and forms grooves [33]. Therefore, it should be clarified that the abrasive particle can deform the material in ways that lead to the removal of only part of the material displaced from the groove.

## 4. Calibration of Wear Coefficient

The pin-on-disc calibration consists of the establishment of the simulation setup, mesh size sensitivity and the wear result analysis, including normal force analysis, wear depth, wear width and wear volume evaluation.

### 4.1. Simulation Setup

The pin-on-disc modelling setup is simplified, as shown in Figure 5. The disc with the radius of 24 mm is meshed by unstructured triangular meshes with specific size. A particle is located at the disc with a distance of 22 mm from the centre of the disc. A cylindrical holder is used to restrict the movement of the particle in horizontal directions. The normal force of 5 N on the particle is generated by applying a particle body force (PBC) through a EDEM API model [23].

Table 2 lists the simulation parameters. The simulation conditions are consistent with the test conditions, except the wear coefficient and rotational speed, which are set as and 5 · 10^−11^ Pa^−1^ and 180 deg/s in the reference simulations, as the different rotational velocity together with a 20% Rayleigh time step lead to stable simulation outcomes.

### 4.2. Mesh Size Sensitivity

Before conducting the mesh size sensitivity analysis, it is essential to clarify the establishment of the wear contours and calculation of the wear volume. To analyse the wear depth and wear width distribution over the wear path, as shown in Figure 6 the disc is divided into 360 subparts, which correspond to 360 wear contours. The establishment of each contour includes 3 steps. First, the disc is reconstructed based on the coordinates of mesh elements. Second, the disc is sliced into 360 subparts with Δ*θ* = 0.5 deg in radian direction of the disc. Third, the coordinates of mesh elements at each subpart are sorted from inner side to outer side of the disc and the sorted coordinates finally form a contour at each position. The contour depth is defined as the maximum displacement of meshes in vertical direction in each wear contour, and the wear width is defined as the width of the opening of each wear contour.

A range of mesh size from 0.1 mm to 1 mm with an increment of 0.1 mm is selected to investigate the effect of the element size on the wear contour, and the wear coefficient is set as 5 × 10^−11^ m^2^/N as a reference. Figure 7 shows the statistical analysis of the wear depth and width of the 360 contours for different mesh size from 0.1 mm to 1 mm, and the error bar denotes the standard deviation. It can be seen that the contour depth decreases with the increase in mesh size, while the wear width shows the opposite trend and increases with the increase in mesh size.

Figure 8 shows an example of wear contours after one revolution. To make it more visible, the figure lists six wear contours from 0.1 mm to 1 mm. Figure 8 indicates that the smaller the mesh size, the narrower and deeper the contour. The reason is that the small meshes indicate a fine meshed surface, and the contact between particle and disc involves more meshes as the contact area is constant, so the deformation of the disc influences more meshes and generates a more precise contour. As the test results shown in Figure 4 indicate, the width of wear contour is around 0.8 mm, to obtain a more precise wear contour, the mesh size is set as 0.1 mm for the following analysis.

### 4.3. Calibration Analysis

Ten revolutions are used to analyse the wear results of the calibration process, including normal force, wear depth, wear width and wear volume. Figure 9 shows the summary of the distribution of normal force between the particle and disc. The box shows the middle portion of the normal force (first quartile to third quartile). Although the normal forces of the 10 revolutions show different distributions from minimum to maximum values, the mean values are all close to 5 N, as represented by the green marker, which means the cumulative force of each revolution is nearly constant. In addition, the normal force contains outliers with high values represented by the circlers. The outliers are caused by the deformation of meshes as the deformed meshes influence the contact with particles at the following time steps.

Figure 10 and Figure 11 illustrate the wear depth and wear width distribution of the 360 wear contours of revolution 10, respectively. It can be seen that wear depth and wear width are randomly distributed at a relatively narrow range. The wear depth is at a range from 0.042 mm to 0.07 mm, and the majority of the contours have depths from around 0.05 mm to 0.065 mm. For the wear width distribution, it has a range from 0.8 mm to 1.6 mm and is mainly distributed at a range of 1–1.5 mm.

To better understand the distribution of wear depth and wear width, the total 10 revolutions are summarised together based on a normality analysis as shown in Figure 12 and Figure 13. The Chi-squared test is used to test the normality of the wear depth and wear width distribution based on a 95% confidence interval. The corresponding *p*-value is summarised in Table 3. For both the wear depth and wear width, one outlier appears at the first four and two revolutions with the maximum value as shown in Figure 12 and Figure 13 and the corresponding *p*-values are close to 0. The reason is that the initial indentation of the particle causes severe deformation which therefore leads to extreme wear depth and wear width. Starting from revolution 8, the *p*-value is higher than 0.05 for both the distributions of wear depth and width, which means the wear depth and wear width follow a normal distribution from revolution 8, so it is considered the simulation is at a run-in phase at the first 7 revolutions. This is because with the continuity of the deformation of the surface, the interconnected common nodes have the tendency to generate a smoothed wear pattern after multiple contacts between particle and meshes [25]. It should be noted that the wear depth increases linearly from R1 to R10 as shown in Figure 12 while the increase rate of wear width becomes slow gradually. This phenomenon can be illustrated by the analysis of the wear contour as shown in Figure 14.

Figure 14 shows 24 contours of the last three revolutions at 8 random positions. For the 8 positions, it can be seen that the wear contours extend both in vertical and horizontal directions with the continuation of the revolution. The wear depth and wear width of R10 are at ranges 0.046–0.057 mm and 1.0–1.5 mm, respectively. For each position, the three wear depths behave similarly from revolution 8 to revolution 10. This is because the relative positions of mesh elements are relatively fixed as a result of the interconnection among nodes, so the deformation of each contour follows a consistent pattern. However, the change in the wear width is not obvious, as the structure of the wear contour as a whole extends in a horizontal direction, and this means the relation between the number of revolutions and the cross-sectional area or wear volume should be described by a nonlinear relation.

Figure 15 shows the relationship between cumulative wear volume and amount of revolutions corresponding to Equations (14)–(17), and the wear volume is calculated on the basis of mesh element, as denoted in Equation (6). Equations (14)–(17) indicate that the wear volume has a quadratic relation with the wear revolution (wear length). This can be explained by analysing the normal force over the disc before and after deformation. For the original surface, as shown in Figure 16a, the normal force on the original surface exists only in the vertical direction. After deformation, as shown in Figure 16b, the normal force has a portion in horizontal direction and the horizontal portion generates no influence on the total normal force between particle and disc. As indicated in Figure 9 the normal force of each revolution is close to a constant, so the increased wear volume is caused by the horizontal portion. With the wear contour getting wider, the increasing trend of the normal force is enhanced and therefore generates more wear.
*y* = 0.012*x* + 0.0008*x*^2^ + 0.001, *R*^2^ = 0.996, (*α_s_* = 1 × 10^−11^ Pa^−1^)
(13)

*y* = 0.023*x* + 0.001*x*^2^ + 0.006, *R*^2^ = 0.995, (*α_s_* = 1.5 × 10^−11^ Pa^−1^)
(14)

*y* = 0.03*x* + 0.001*x*^2^ + 0.006, *R*^2^ = 0.995, (*α_s_* = 1.7 × 10^−11^ Pa^−1^)
(15)

*y* = 0.042*x* + 0.0012*x*^2^ + 0.03, *R*^2^ = 0.997, (*α_s_* = 2 × 10^−11^ Pa^−1^)
(16)

*y* = 0.098*x* + 0.011*x*^2^ + 0.08, *R*^2^ = 0.997, (*α_s_* = 5 × 10^−11^ Pa^−1^)
(17)


As presented above, the pin-on-disc numerical model needs 8 revolutions (run-in period) to reach a steady state, so the total 1302.5 revolutions of the test are determined to be modelled with 12 revolutions. As shown in Figure 15, the wear coefficient is changed systematically, and finally, the value of $1.7\times 10^{-11}\;Pa^{-1}$ can reach the range of the test volume. It should be noted that with the increase in the wear coefficient, the relation between wear volume and the number of revolutions presents a more obvious accelerating tendency. This is because a higher wear coefficient means a larger surface deformation, and the deformed surface in turn increases the normal force in horizontal direction as shown in Figure 16, so that the wear volume of each revolution increases faster.

Furthermore, the other wear coefficient lower than $2\times 10^{-11}\;Pa^{-1}$ can also be selected when the different revolutions are chosen to represent the total test revolutions. For example, $1\times 10^{-11}\;Pa^{-1}$ can be applied when the number of revolutions determined is higher than 12. In total, 9–11 revolutions can also represent the total test revolutions when $2\times 10^{-11}\;Pa^{-1}$ is selected. Therefore, the scaling factor, which is defined as the ratio of the number of test revolutions to that of the modelling revolutions, depends on the match of the amount of revolution and the wear coefficient.

## 5. Verification of Numerical Results

After calibration of the wear coefficient, the simulation results can be verified by comparing the test results, including wear depth and width of the wear contour, wear volume and wear profile. The corresponding simulation and test results are summarised in Table 4. The discrepancy is calculated as the ratio of the absolute difference between the numerical and test results to the test result. The wear depth and wear width of the test results are obtained based on the previous study [4], and the simulation results are based on the statistical analysis of 360 contours as explained in Figure 6. It can be seen that the numerical wear volume is close to the test result, with a discrepancy of 6%.

For the average wear depth and wear width, the numerical results show 36% and 34% discrepancy, respectively. The difference can be explained by four factors [31]: wear volume calculation in model and experiment, wear mechanism and number of samples from experiments and wear of particle tip in the experiment. First, the calculation of the wear volume is different. For the test, the wear mass is obtained by calculating the wear mass loss. This means the wear volume only counts the dispersed portion. However, the wear process also leads to plastic deformation, and the plastic deformed portion directly affects the wear profile. For the numerical model, the wear volume is directly calculated from the wear profile. Second, the wear mechanisms are different. For the test, the geometry is mainly deformed by micro-cutting and micro-ploughing (plastic deformation), so a portion of the deformed geometry generates wear loss. This is why the ridges are formed as shown in Figure 4c. For the numerical model, the deformation of the meshes only occurs in the normal direction of mesh element and all the deformation is considered as wear loss. Third, the number of samples for the calculation of wear depth and width has enormous difference. The test results of the wear depth and wear width are only based on three contours and the randomness of the sample selection causes deviation when compared with the whole wear profile. However, 360 wear contours are extracted from the numerical model. Fourth, the tip of the iron ore in the test is worn off and therefore the contact area changes, so the corresponding wear width and depth changes. This situation is avoided in the simulations.

Figure 17 presents the reconstructed deformed disc. It can be seen that the sliding of the particle forms a relatively smooth profile around the disc, except for several highly deformed spots ranging from 90–120°. The reason is that the normal force at each revolution appears to be high value, as shown in Figure 9 and these normal forces lead to extreme deformation. For the cross-section of the wear profile at each degree, the wear depth has the highest value at the middle and decreases toward the two sides gradually. Figure 17 shows that an extremely deformed contour presents at the original position of the disc, which is because the initial indentation of the particle causes the severe deformation. This is reasonable, as this extreme deformation verifies the appearance of a outlier in the distribution of wear depth and wear area, as shown in Figure 12 and Figure 13.

To compare the wear contour with the test, three wear contours are extracted randomly from the wear profile, as shown in Figure 18. Figure 18 indicates that the width and depth of the contours are higher and deeper than those of the test contours, while the contours are similar in shape. The results show that by properly scaling up the wear coefficient and selecting the mesh size, the long-term or long-distance laboratory test can be modelled by a short-term or short-distance numerical model with an acceptable deviation.

## 6. Conclusions

A standard sliding wear process, using a pin-on-disc test, is investigated in order to indicate that a long-term or long-distance laboratory test can be modelled by a short-term or short-distance numerical model. This numerical model is built by combining the Archard wear model with a deformable geometry technique. The wear results, including wear contour and wear volume, are evaluated statistically with different mesh sizes. Three wear properties, i.e., wear depth, wear width and wear volume, are compared with the test results.

For the sliding of a particle with the radius of 3 mm, the mesh size of the disc is set as 0.1 mm. The wear contour indicates that the mesh size from 0.1 mm to 1 mm has a significant effect on the wear contour. The coarser mesh generates wider and shallower contours, while a fine mesh can obtain comparable wear contour with test results. This is because the contact between particle and disc involves more meshes with the fine meshed surface, so the deformation of the disc influences more meshes and generates a more precise contour.

For the geometrical deformation technique, the wear path reaches a relatively steady deformation state after eight revolutions. It is verified that the wear depths and wear widths of the wear contours follow normal distributions after a run-in phase because of the self-smoothing effect of the interconnected common nodes of the mesh elements.

The wear coefficient is calibrated by comparing the wear volume between simulation and test results, and the wear volume has a quadratic relation with the number of revolutions. The wear volume of the test accurately predicts the simulation results with the minimum number of revolutions when the wear coefficient is lower than $2\times 10^{-11}$. For different wear coefficients, the wear volumes correspond to different quadratic relations. The lower the wear coefficient, the weaker the tendency of the quadratic relation.

To obtain comparable wear results, the minimum number of revolutions is determined as eight, and the corresponding maximum scaling factor, which is defined as the ratio of the number of test revolutions to that of the modelling revolution, is 162.8. By scaling up the wear coefficient and properly selecting the mesh size, the long-term laboratory pin-on-disc test can be modelled by a short-term numerical model. This scaling effect can significantly save computational time and improve efficiency with promising calculation precision.

In this paper, the presented methodology was successfully demonstrated on a single material combination. This methodology can be used for different material combinations. To establish a numerical model for the deformation of other material combinations, the corresponding tests and simulations should be performed. 

## Figures and Tables

**Figure 1 materials-15-01813-f001:**
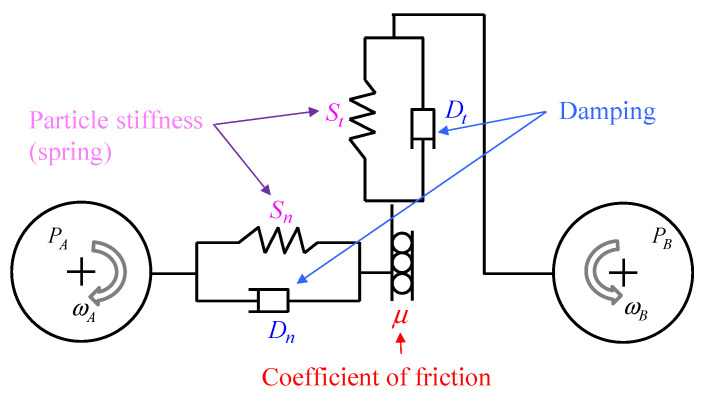
Illustration of contact between two particles.

**Figure 2 materials-15-01813-f002:**
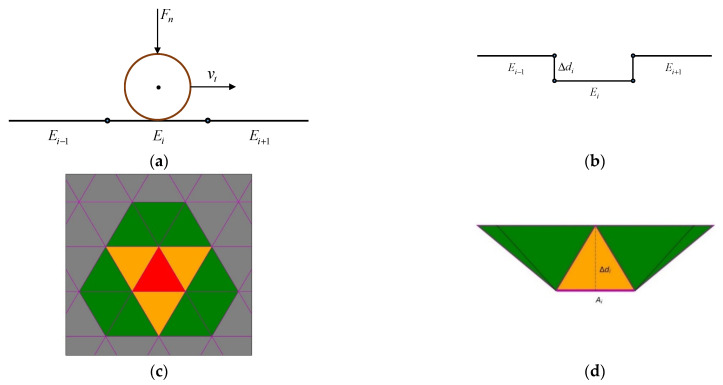
Mesh deformation procedure [25]. (**a**) Particle in contact with the surface Ai of the mesh element; (**b**) displacement of the mesh element; (**c**) side view of the interconnection between nodes and (**d**) top view of the wear representation.

**Figure 3 materials-15-01813-f003:**
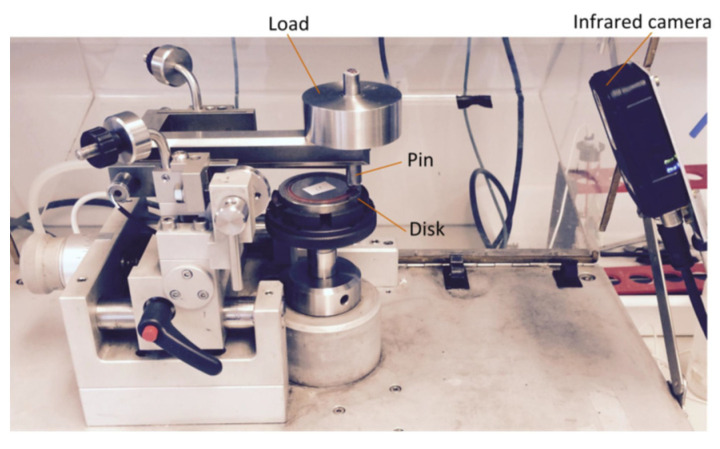
Pin-on-disc apparatus [31].

**Figure 4 materials-15-01813-f004:**
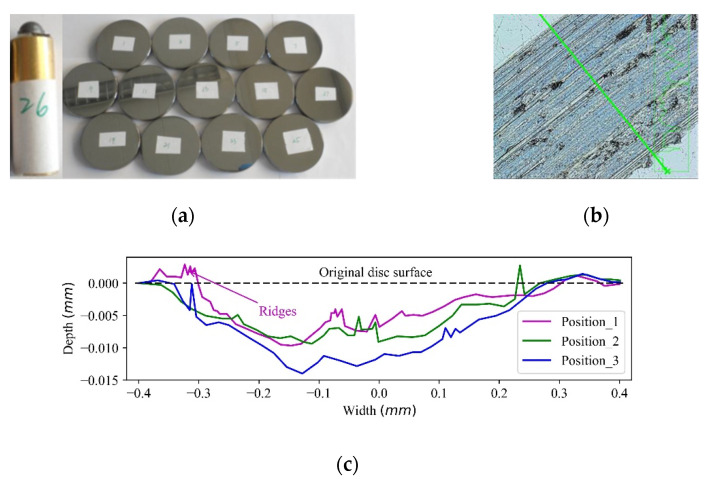
Wear contour extraction. (**a**) An iron ore with a spherical head; (**b**) measuring of the cross-section [31] and (**c**) three wear contours based on the measurement after test.

**Figure 5 materials-15-01813-f005:**
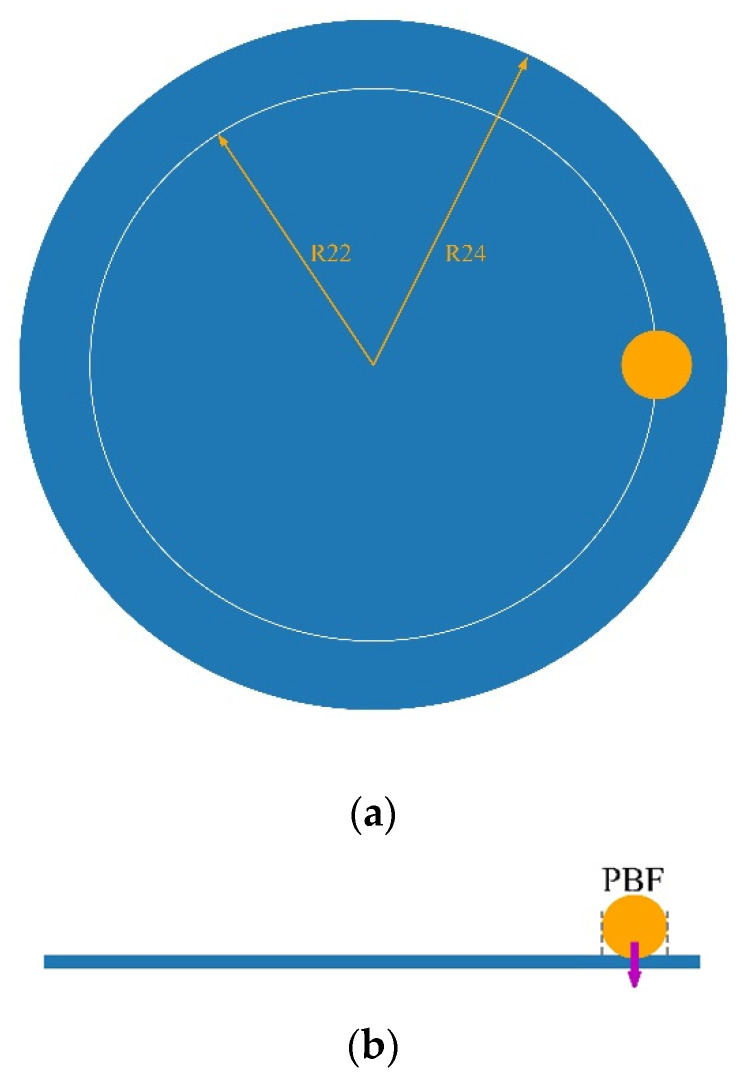
Pin-on-disc setup. (**a**) Top view and (**b**) side view.

**Figure 6 materials-15-01813-f006:**
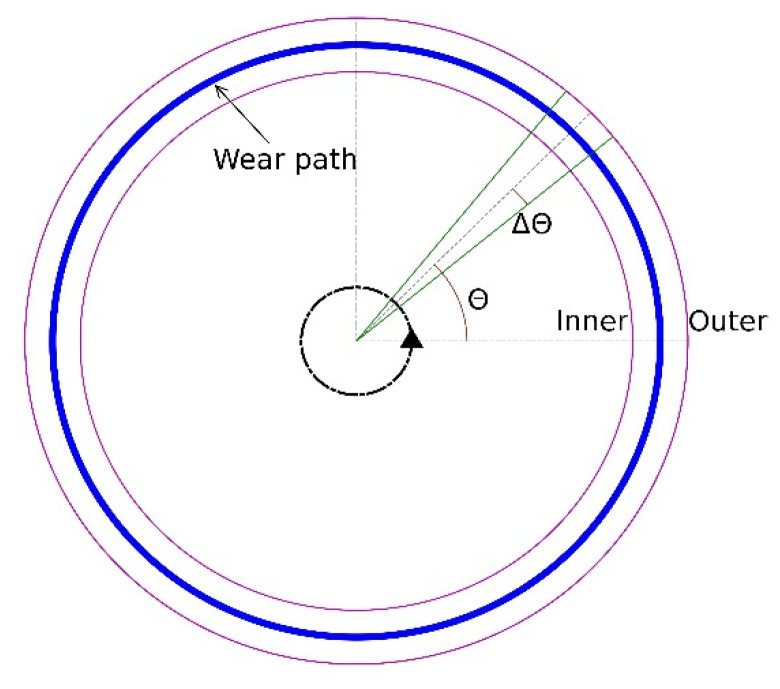
Illustration of data extraction.

**Figure 7 materials-15-01813-f007:**
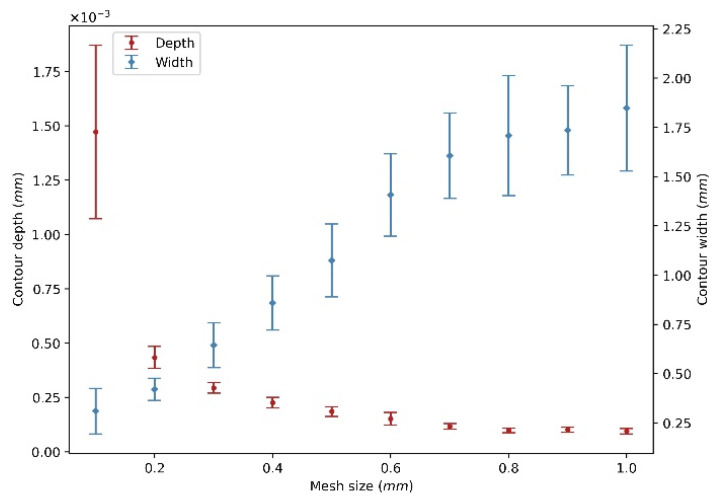
Statistical analysis of wear depth and width.

**Figure 8 materials-15-01813-f008:**
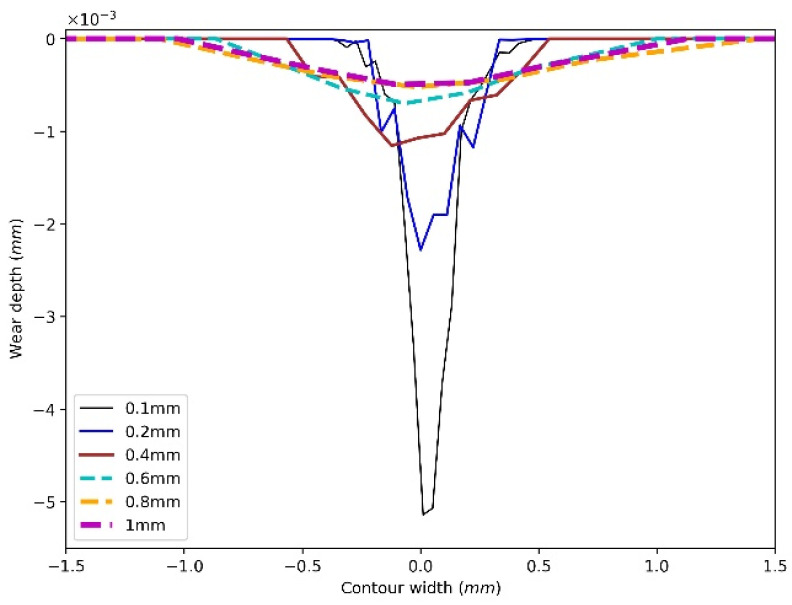
Effect of mesh size on wear contour.

**Figure 9 materials-15-01813-f009:**
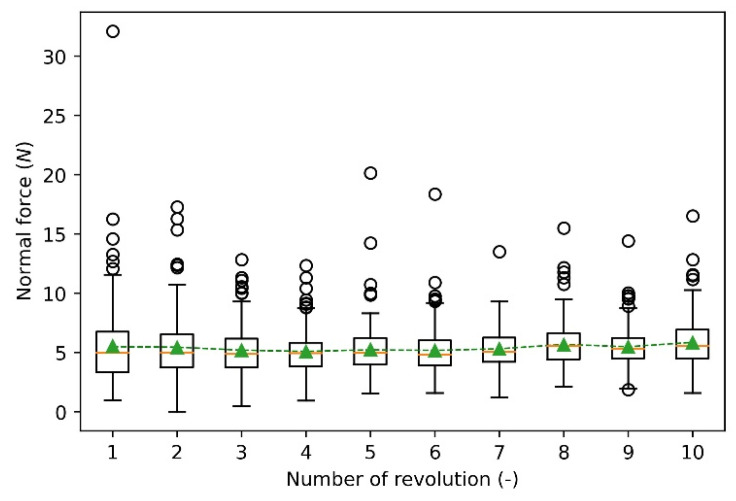
Normal force of ten revolutions.

**Figure 10 materials-15-01813-f010:**
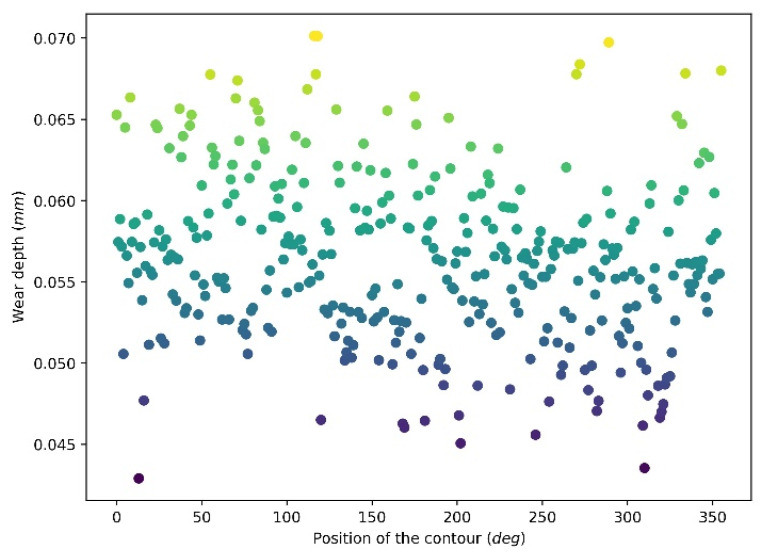
Wear depth distribution after 10 revolutions.

**Figure 11 materials-15-01813-f011:**
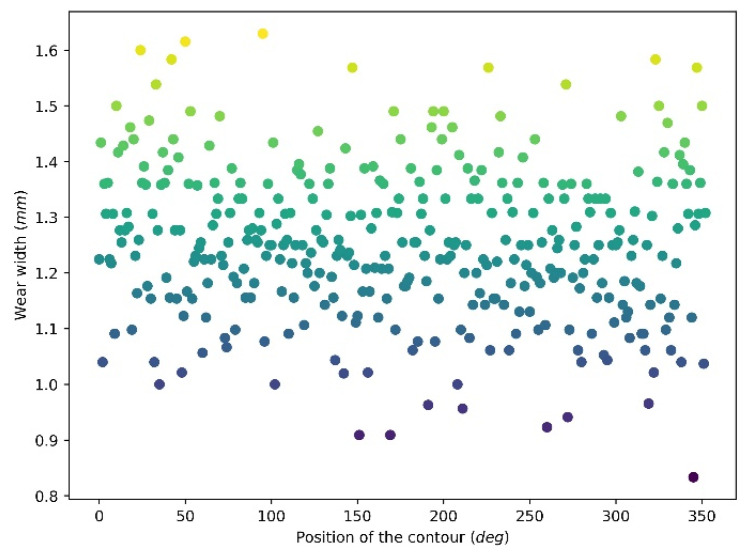
Wear width distribution after 10 revolutions.

**Figure 12 materials-15-01813-f012:**
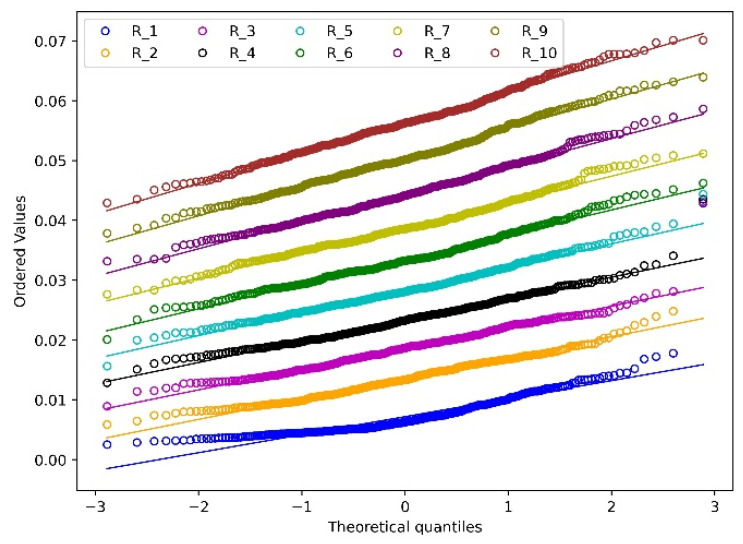
Normality analysis of wear depth.

**Figure 13 materials-15-01813-f013:**
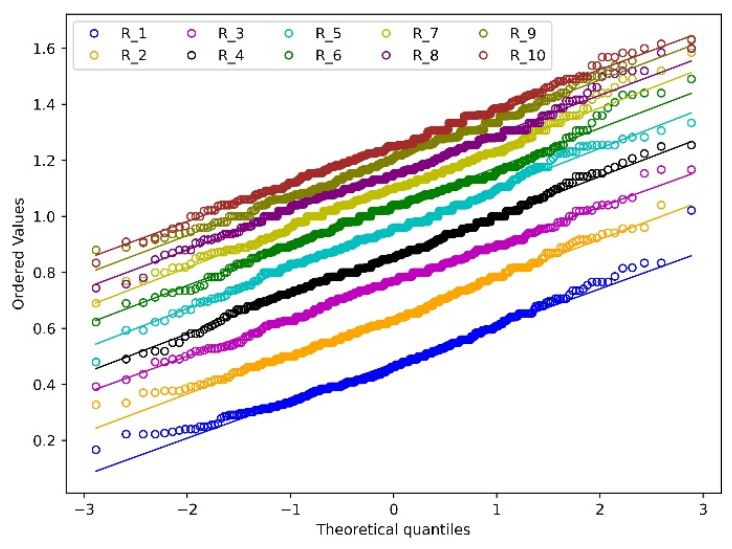
Normality analysis of wear width.

**Figure 14 materials-15-01813-f014:**
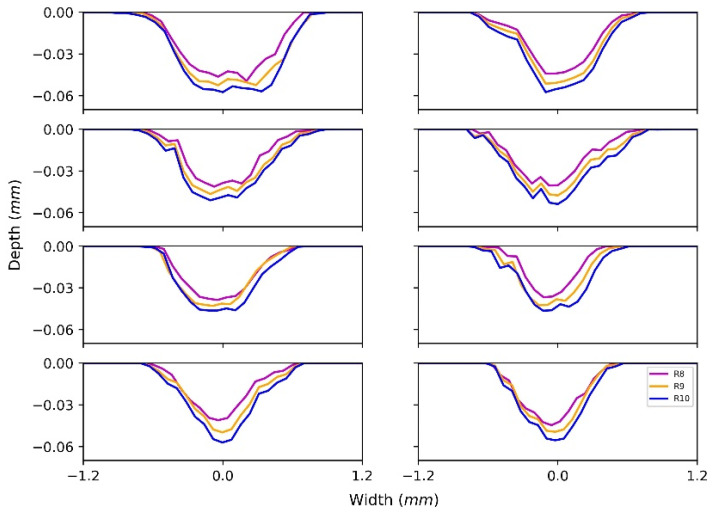
Examples of wear contour.

**Figure 15 materials-15-01813-f015:**
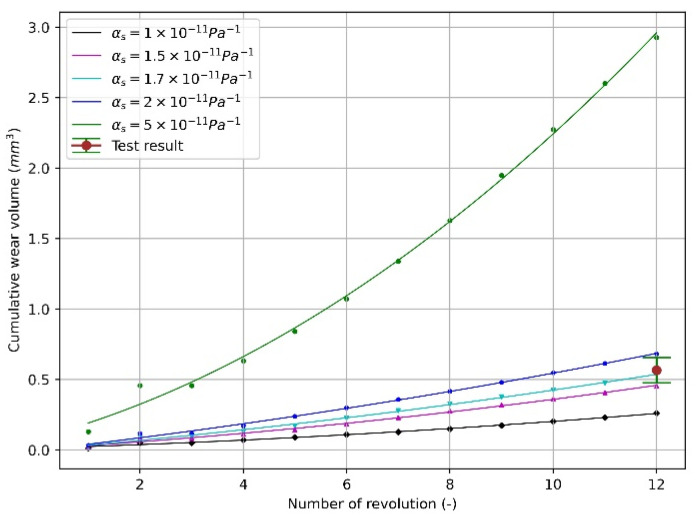
Wear volume as a function of revolution.

**Figure 16 materials-15-01813-f016:**
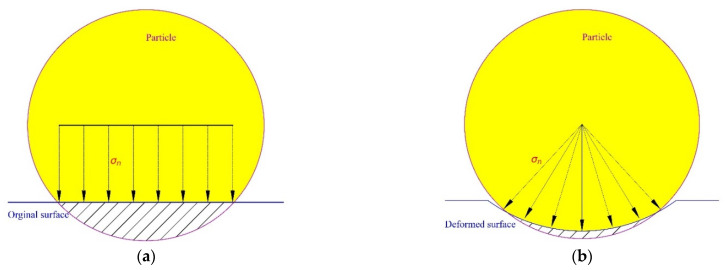
Illustration of stress over disc. (**a**) Original surface and (**b**) deformed surface.

**Figure 17 materials-15-01813-f017:**
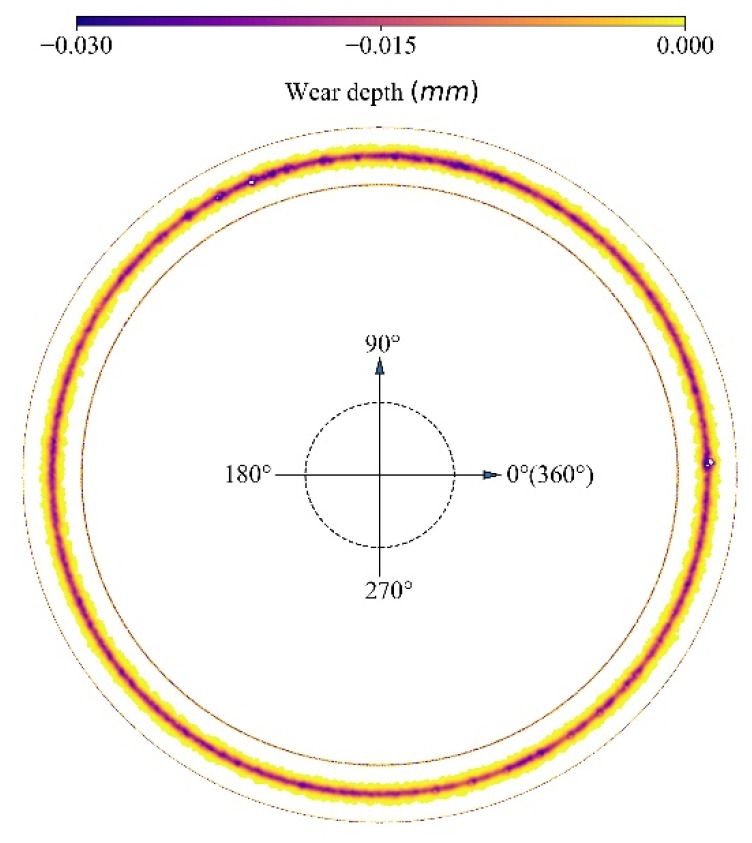
Wear profile.

**Figure 18 materials-15-01813-f018:**
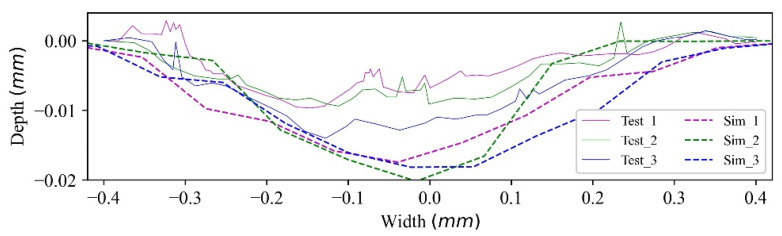
Wear contour comparison.

**Table 1 materials-15-01813-t001:** Parameters of samples and tests.

Categories	Parameters	Values
Iron ore	radius (mm)	3
	hardness (*H_v_*)	476 ± 9
	hardness (*H_m_*)	4–4.5
	density (kg/m^3^)	4850
Mild steel	density (kg/m^3^)	7932
	hardness (*H_v_*)	134 ± 4
Test	indentation force (N)	5
	rotational speed (deg/s)	390.8
	sliding distance (m)	180
	revolutions (-)	1302.5
	rotational radius (mm)	22

**Table 2 materials-15-01813-t002:** Parameters for DEM simulations [34].

Categories	Parameters	Values
Iron ore	radius (mm)	3
	density (kg/m^3^)	4850
	Poisson’s ratio	0.24
	shear modulus (GPa)	0.1
Mild steel	density (kg/m^3^)	7932
	Poisson’s ratio	0.3
	shear modulus (GPa)	78
	mesh size (mm)	0.1
Contact	coefficient of restitution *e*	0.4
	coefficient of static friction *μ_s_*	1.0
	coefficient of rolling friction *μ_r_*	0
Conditions	indentation force (N)	5
	rotational speed (deg/s)	180
	rotating radius (mm)	22
	coefficient of sliding wear *α_s_*(×10^−11^ Pa^−1^)	5
	time step Δ*t*(×10^−5^ s)	1.4

**Table 3 materials-15-01813-t003:** Summary of *p*-value based on Chi-squared test.

	R 1	R 2	R 3	R 4	R 5	R 6	R 7	R 8	R 9	R 10
Wear depth	0	0	0	0	1.3 × 10^−12^	0.03	0.01	0.11	0.33	0.54
Wear width	0	0	0.001	0.02	0.04	0.001	0.02	0.21	0.29	0.15

**Table 4 materials-15-01813-t004:** Summary of numerical and test results.

	Wear Depth (mm)	Wear Width (mm)	Wear Volume (mm^3^)
Test result	0.014	0.8	0.565
Numerical result	0.019	1.07	0.532
Discrepancy	36%	34%	6%

## Data Availability

Not applicable.

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
