# Peer review of "Pin-on-Disc Modelling with Mesh Deformation Using Discrete Element Method"

_materials, 2022, doi:10.3390/ma15051813_

Round 1
Reviewer 1 Report
Include units for Equation 1 and components in that equation.
Section 5, verification of numerical results is not convincing. The discrepancy/ error between test result and numerical result is as high as 34-36% which is very high to accept the numerical modeling.
The reasons cited for this discrepancy "the test results of the wear depth and wear width are only based on three contours and the randomness of the sample selection causes deviation when comparing with the whole wear profile......., need to be justified with citation/references/comparison etc.
The overall results also applicable to only a particular material combination with high discrepancy. Further improvement on this suggested.
Reviewer 2 Report
The paper is very interesting, it is written very well.
I can recommend only small changes that could improve its quality:
1) Please move the section describing pin-on-disk tests and materials in the tribopair (lines 142 - 161) to the Materials and methods section.
2) Please provide the elemental composition of the steel or at least its grade.
3) Please add Mohs hardness of the iron ore to Table 1 if possible because this scale is mostly used for minerals.
Round 2
Reviewer 1 Report
The revised manuscript is satisfactory.